# Reliability-enhanced data cleaning in biomedical machine learning using inductive conformal prediction

**Xianghao Zhan** [1,2], **Qinmei Xu** [2], **Yuanning Zheng** [2], **Guangming Lu** [3], **Olivier Gevaert** [2]*

**1** Department of Bioengineering, Stanford University, Stanford, California, United States of America, **2** Department of Biomedical Data Science, Stanford University, Stanford, California, United States of America, **3** Department of Medical Imaging, Jinling Hospital, Nanjing, Jiangsu, People's Republic of China

* ogevaert@stanford.edu

**Data availability statement:** The drug-induced liver injury dataset is publicly at:

## Abstract

Accurately labeling large datasets is important for biomedical machine learning yet challenging while modern data augmentation methods may generate noise in the training data, which may deteriorate machine learning model performance. Existing approaches addressing noisy training data typically rely on strict modeling assumptions, classification models and well-curated dataset. To address these, we propose a novel reliability-based training-data-cleaning method employing inductive conformal prediction (ICP). This method uses a small set of well-curated training data and leverages ICP-calculated reliability metrics to selectively correct mislabeled data and outliers within vast quantities of noisy training data. The efficacy is validated across three classification tasks with distinct modalities: filtering drug-induced-liver-injury (DILI) literature with free-text title and abstract, predicting ICU admission of COVID-19 patients through CT radiomics and electronic health records, and subtyping breast cancer using RNA-sequencing data. Varying levels of noise to the training labels were introduced via label permutation. Our training-data-cleaning method significantly enhanced the downstream classification performance (paired t-tests, $p \leq 0.05$ among 30 random train/test partitions): significant accuracy enhancement in 86 out of 96 DILI experiments (up to 11.4% increase from 0.812 to 0.905), significant AUROC and AUPRC enhancements in all 48 COVID-19 experiments (up to 23.8% increase from 0.597 to 0.739 for AUROC, and 69.8% increase from 0.183 to 0.311 for AUPRC), and significant accuracy and macro-average F1-score improvements in 47 out of 48 RNA-sequencing experiments (up to 74.6% increase from 0.351 to 0.613 for accuracy, and 89.0% increase from 0.267 to 0.505 for F1-score). The improvement can be both statistically and clinically significant for information retrieval, disease diagnosis and prognosis. The method offers the potential to substantially boost classification performance in biomedical machine learning tasks without necessitating an excessive volume of well-curated training data or strong data distribution and modeling assumptions in existing semi-supervised learning methods.

https://bipress.boku.ac.at/camda2022/. The
RNA-sequencing dataset for breast cancer
subtyping is publicly available at:
https://www.nature.com/articles/nature11412.
The code implementation of the methods is
available at:
https://github.com/xzhan96-stf/icp_train_clean.
This study utilized COVID-19 dataset described
in the following publication: Xu, Qinmei,
Xianghao Zhan, Zhen Zhou, Yiheng Li, Peiyi Xie,
Shu Zhang, Xiuli Li et al. "AI-based analysis of
CT images for rapid triage of COVID-19
patients."
https://doi.org/10.1038/s41746-021-00446-z.
Interested researchers can submit their data
access requests, along with inquiries regarding
inclusion and exclusion criteria, through the
following contact details: Ethics Committee of
Jinling Hospital, Nanjing University Affiliated,
Phone: +86-025-80863234, Email:
xqm@smail.nju.edu.cn.

**Funding:** The research reported here was
further supported by the National Cancer
Institute (NCI) (R01CA260271 to OG). The
funders had no role in study design, data
collection and analysis, decision to publish, or
preparation of the manuscript.

## Author summary

In biomedical machine learning, noisy training data often compromise the performance
of models critical for clinical decision-making. Generating well-curated datasets is chal-
lenging, while noisy datasets are prevalent, especially with advanced data augmentation
techniques. This study introduces a novel reliability-based training data-cleaning method
employing inductive conformal prediction (ICP). Using a small, well-curated calibration
set, the method identifies and corrects mislabeled samples and removes outliers, enhanc-
ing label quality without strong assumptions on data distribution or model structure.
We validated the approach across three diverse tasks: filtering drug-induced liver injury
(DILI) literature, predicting ICU admissions of COVID-19 patients from radiomics and
clinical data, and subtyping breast cancer based on RNA-seq profiles. Results showed sig-
nificant improvements in classification performance, even under varying levels of label
noise. This method demonstrates a practical solution for leveraging large, noisy datasets
in biomedical applications, reducing reliance on extensive manual labeling, and improv-
ing the reliability of machine-learning models across modalities. Our findings highlight
the potential of ICP to advance data-cleaning strategies in noisy real-world settings.

# 1. Introduction

## 1.1. Challenge of well-curated labeled datasets in biomedical machine learning

Machine learning (ML), especially in supervised learning, has become foundational in a wide
range of biomedical applications. Recent advances have demonstrated the potential of ML
models trained on multi-modal and multi-omics biomedical datasets to support critical tasks
such as clinical decision support [1], patient stratification [2], information retrieval [3] and
outcome prediction [4,5]. In supervised learning, acquiring accurately labeled training data
is critical for developing effective models. However, with the growing amount of multi-modal
data of images, genomics, clinical records and unstructured narratives, curating high-fidelity,
accurately labeled datasets is challenging. In biomedical image analysis, for instance, train-
ing an effective supervised learning model often requires experienced radiologists annotating
thousands of radiological images (e.g., X-rays, CT scans) to ensure accuracy [2,6]. Similarly,
to train an effective model for biomedical natural language processing, experts need to read
through thousands of free-text notes to generate text classifications [7]. This intensive manual
annotation is prone to human error, introducing labeling noise as a common issue. Moreover,
modern ML data augmentation techniques, such as weak labeling [8–10], interpolation-based
upsampling (e.g., synthetic minority oversampling technique (SMOTE) [2]), can also intro-
duce noise. While these approaches expand training data availability, the resulting noise may
undermine model performance, potentially leading to misdiagnoses and reduced prediction
reliability [11–14]

## 1.2. Literature review and research gaps

To mitigate the adverse impacts on model performances caused by the noises in the labels (i.e.
wrongly labeled data and outliers), existing approaches often rely on strict assumptions about
data distribution, prior knowledge on noise patterns, training separate classification models,
which may under-utilize noisy labels, and lack uncertainty information. These methods can
be broadly classified into two categories: 1) leverage noisily labeled data in the model training

with specific strategies to correct/weigh noisy labels [14–19] or 2) perform semi-supervised learning by treating the noisy labeled data as unlabeled data while only using the features [20–22]. In the first category, researchers typically estimate the noise transition matrix [15,16,23], make loss corrections in ML model development and perform re-weighting among the accurately labeled data (clean data) and noisily labeled data [19,24], or make corrections of the noisy labeled data based on ML model predictions [14,17,25]. However, these methods are limited in several ways: 1) estimating the noise transition matrix is quite challenging without prior knowledge of the noise patterns [14–16]; 2) the methods weighting between the noisy data and clean data have the challenging issue to design reliable criteria to filter the clean data, and can suffer from the confirmation bias if the noisy data are wrongly treated as the clean data [14,24]; 3) the methods that try to correct the noisy training data rely heavily on the assumptions of the classification models, and typically do not perform well under highly noisy circumstances, and can overfit the clean data [14,17,18,25]; 4) the label-correction models typically lacks statistical calibration and do not provide or leverage any prediction reliability and uncertainty information.

In the second category, researchers treat the noisy data as unlabeled data and add additional terms to the learning objectives and loss functions [20–22]. The key limitations of the semi-supervised learning approaches are 1) the assumptions on the data distribution (e.g., the unlabeled data distribute on the same manifold as the labeled data) [20–22,26] and 2) under-utilize the potential information contained in the noisy labels.

## 1.3. Contribution of this study

To tackle these challenges, we introduce a reliability-based data cleaning method using inductive conformal prediction (ICP) to identify and selectively correct noisy labels in biomedical datasets. Our statistical approach detects mislabeled data, where another label may be more accurate, as well as outliers that likely do not belong to any label. We validated the method's effectiveness on three biomedical classification tasks, covering clinical natural language processing (NLP), multi-modal image and electronic health record data, and RNA sequencing. To simulate real-world noise, we introduced varying levels of label contamination and assessed classification performance with and without data cleaning. Additionally, we analyzed the cleaning process, tracking corrections and outlier removals under different hyperparameter settings.

This study uniquely combines ICP with a training data cleaning approach that identifies mislabeled samples and outliers, improving the reliability of the ML label in noisy biomedical datasets. By leveraging a small, high-quality calibration set, this method achieves significant classification improvements without assuming a fully clean dataset, addressing practical constraints in biomedical data curation.

The key contributions of our proposed approach that differentiates itself from existing approaches are:

- **Adaptive to multi-modal datasets**: we systematically and statistically validated the effectiveness in cleaning noisy training data and significantly improving classification performance under various noise levels, on three biomedical machine learning tasks, and across different data modalities.
- **Less strict modeling assumption**: the ICP framework rests on a more relaxed assumption on the independently and identically data distribution [27–29];

- **Less reliance on well-curated data and less overfitting risk**: the methodology decouples the noisy data and clean data by separate steps to compute the nonconformity measure (with a proper training set) and to quantify the uncertainty by calibration (with a calibration set), which reduces the label-correction-process's reliance on the clean data and mitigate the problem of overfitting the clean data;
- **Less reliance on classification models**: the ICP-based data cleaning method directly leverages the prediction uncertainty information via statistical calibration on the calibration set, instead of being solely based on classifier's prediction. The statistical calibration reduces the dependency on the modeling assumptions similar to previous work that uses the label correction process on the specific models' predictions [14,17,18];
- **Better utilization of noisy labels**: the method leverages the label information of the noisy dataset instead of completely deeming them as unlabeled data as in traditional semi-supervised learning [20–22].

## 2. Methods

### 2.1. Datasets and task description

To investigate a broad range of medical data modalities and different scales of feature dimensionality, in this study, we tested the ICP-based training data cleaning method on three classification tasks: 1) a natural language processing task: filter drug-induced liver injury (DILI) literature based on word2vec (W2V) and sent2vec (S2V) embeddings [7]; 2) an imaging and electronic health record task: predict whether a COVID-19 patient in the general ward will be admitted to the intensive care unit (ICU) [2]; 3) an RNA-sequencing (RNA-seq) task: classify breast cancer subtypes based on The Cancer Genome Atlas Program (TCGA) RNA-seq dataset [30]. The details of these datasets are introduced in Table 1 and the following paragraphs.

The DILI dataset was released by the Annual International Conference on Critical Assessment of Massive Data Analysis (CAMDA 2021), with the DILI-positive samples (7,177) and DILI-negative samples (7,026) curated by FDA experts. The data involve both title and abstract of publications and the task is to predict whether a publication contains DILI information. The detailed pre-processing of the text can be found in the previous publication [7]: the text was lower-cased, with additional removals of punctuation, numeric, special characters, multiple white spaces, stop words, and the text was finally tokenized with Gensim library on Python 3.7 [31]. Then, we generated the text embeddings based on the biomedical sent2vec (S2V) model [32] and biomedical word2vec (W2V) model [33] because they

**Table 1. The basic information of datasets used in this study.**

| Dataset | Modality | Task type | Features | Dimensionality | # of samples | Metrics | Ref. |
|---------|----------|-----------|----------|----------------|--------------|---------|------|
| DILI | Text | Binary | W2V, S2V | 200, 700 | 14,203 | Accuracy | [7] |
| | | | | | (+: 7,177, -: 7,026) | | |
| COVID-19 | Imaging, EHR | Binary | Radiomics, | 9,940 | 2,113 | AUROC, AUPRC | [2] |
| | | | lab and clinical data | | (+: 60, -: 2,053) | | |
| TCGA | RNA-seq | Multi-class | RNA-seq | 31,098 | 1,068 | Accuracy, F1 score | [30] |

have shown good classification performances on this challenge in previous studies with logistic regression classifiers, and these two text vectorizations generate text embeddings with low feature dimensionality (700 for S2V and 200 for W2V) [7].

COVID-19 dataset (n=2,113) were collected from 40 hospitals in China from December 27, 2019 to March 31, 2020 [2]. Patient selection followed the inclusion criteria: (a) RT-PCR confirmed positive severe acute respiratory syndrome coronavirus (SARS-CoV-2) nucleic acid test; (b) baseline chest CT examinations and laboratory tests on admission; (c) short-term prognosis information (discharge or admission to ICU). Data for each patient included: 1) Clinical data based on electronic Health Records (EHR): (a) demographics: age and gender; (b) comorbidities: coronary heart disease, diabetes, hypertension, chronic obstructive lung disease (COPD), chronic liver disease, chronic kidney disease, and carcinoma; (c) clinical symptoms: fever, cough, myalgia, fatigue, headache, nausea or vomiting, diarrhea, abdominal pain, and dyspnea on admission. 2) Lab data based on laboratory test: (a) blood routine: white blood cell (WBC) count ($\times 10^9/L$), neutrophil count ($\times 10^9/L$), lymphocyte count ($\times 10^9/L$), platelet count ($\times 10^9/L$), and hemoglobin ($g/L$); (b) coagulation function: prothrombin time (PT) ($s$), activated partial thromboplastin time (aPTT) ($s$), and D-dimer ($mg/L$); (c) blood biochemistry: albumin ($g/L$), alanine aminotransferase (ALT) ($U/L$), aspartate Aminotransferase (AST) ($U/L$), total bilirubin ($mmol/L$), serum potassium ($mmol/L$), sodium ($mmol/L$), creatinine ($\mu mol/L$), creatine kinase (CK) ($U/L$), lactate dehydrogenase (LDH) ($U/L$), $\alpha$-Hydroxybutyrate dehydrogenase (HBDH) ($U/L$); (d) infection-related biomarkers: C-reactive protein (CRP) ($mg/L$). 3) Radiomics data based on CT imaging: a commercial deep-learning AI system (Beijing Deepwise & League of PhD Technology Co. Ltd) was first used to detect and segment the pneumonia lesion, and two radiologists checked the results of the automatic segmentation. Then, pyradiomics (v3.0) running in the Linux platform was adopted to extract radiomic features (1,652 features per lesion). Next, for a given patient and each radiomic feature, we summarized the distribution of the feature values across all the lesions for the patient by several summary statistics (mean, median, standard deviation, skewness, the first quartile, the third quartile) and the number of lesions. Finally, a total of 9,913 quantitative radiomic features were extracted from CT images for each patient. Detailed data collection and preprocessing can be found in the S1 Text and [2].

RNA-seq data of the TCGA breast cancer cohort [30] were obtained from the UCSC Xena browser [34]. Gene expression values were normalized using the Fragments Per Kilobase of transcript per Million mapped reads (FPKM) method. The dataset included 1,068 patients, and the molecular subtypes were labeled using the methods as described previously [35]. The number in each subtype was: LumA (n = 555), LumB (n = 210), Basal (n = 184), HER2 (n = 81), and normal-like (n = 38). The feature dimensionality is 31,098.

## 2.2. Inductive conformal prediction: a building block for training data cleaning

This section presents the inductive conformal prediction-based method utilized for identifying and correcting noisy labels (Fig 1), where the calibration phase quantifies label reliability across samples, serving as the basis for data cleaning (step 1 in Fig 1).

**2.2.1. General pipeline of inductive conformal prediction** Inductive Conformal Prediction (ICP) is a computational framework that operates under the assumption of identically and independently distributed data. For comprehensive discussions on conformal predictors, readers are directed to past works [27,28,36]. To briefly describe the ICP framework, it begins with splitting the training data into a proper training set and a calibration set. Non-conformity measures (also referred to as conformity score, nonconformity measurements in

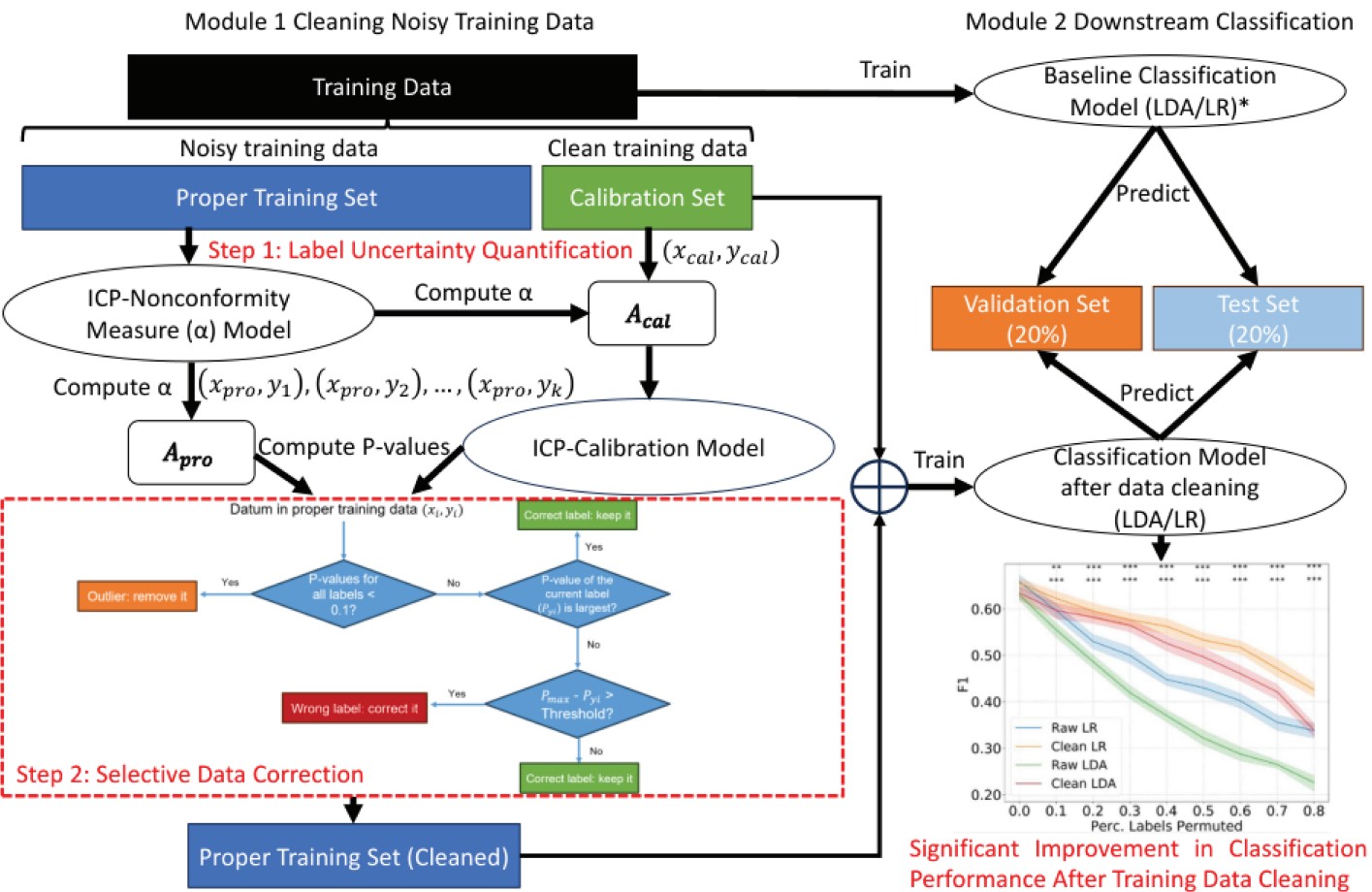

**Fig 1. The design of the reliability-based training data cleaning method based on inductive conformal prediction and the validation process.** The training data cleaning method based on the conformal prediction is shown on the left half (module 1) while the modeling of the downstream classification tasks and the evaluation of the validation and test sets are shown on the right half of the figure (module 2). Based on the standard ICP method, the training dataset is partitioned into the proper training set and calibration set. The proper training set is used to represent the noisy training data and the calibration set represents the well-curated dataset. Wrongly labeled data and outliers in the proper training set are detected and corrected based on the P-values calibrated on the nonconformity measure distribution on the calibration set. The cleaned training set is then used to train classifiers for downstream classification tasks and compared against baselines.

different studies) are then derived from the proper training set using specific heuristic rules or algorithms, such as using the conditional probability from a classification model like in the Conformal Prediction with Shrunken Centroids (CPSC) [37] or using the ratio of cumulative distances between dissimilar and similar samples as in Conformal Prediction with k-nearest neighbors (CPKNN) [29,36]. Following this, nonconformity measures for the calibration and test data are computed and utilized as calibration statistics. For the calibration set, these nonconformity measures are computed using both features and ground-truth labels. In contrast, for the test data, the nonconformity measures are computed for every possible label within the label space in order to quantify and compare the conformity of each possible label. ICP then uses the empirical distribution of the nonconformity measures on the calibration set as a reference, assessing at which percentile the test sample-label combination's nonconformity measure falls. Based on this percentile, ICP calculates the P-value, which indicates the degree of conformity of a specific feature-label combination to the underlying data distribution, i.e.,

how well the test feature-label combination conforms to the distribution of the nonconformity measure on the calibration set. This enables the quantification of prediction reliability by considering the conformity of the most likely label to the training data distribution.

The ICP framework used in this study is visualized in the top-left section of Fig 1, where the entire training set with $b$ samples is divided into a proper training set $\{X_1, X_2, X_3, \ldots, X_a\}$ and a calibration set $\{X_{a+1}, X_{a+2}, X_{a+3}, \ldots, X_b\}$ ($a < b$). Based on a nonconformity measure algorithm trained on the proper training set, the nonconformity measure of the calibration set and proper training set ($A\_Cal$ and $A\_Pro$) are calculated. The P-values, which are the calibrated results reflecting the conformity of a feature-label combination, are then calculated for all possible labels of each sample in the proper training set, and the P-values indicate the reliability of a proper training sample when attached to every possible label judged by the nonconformity measure distribution on the calibration set. In this study, to clean the training data with noisy labels, the proper training data represent the majority part of the labeled data but may be noisy with outliers and wrongly labeled data unknown to the users, while the calibration set is the smaller portion of labeled data with well-curated labels. It should be mentioned that different from the typical use case of ICP where it uses the calibration set to compute the P-values for the test set, in this study, we aim to compute the P-values for the samples in the proper training set and leverage the P-values for training data cleaning.

The nonconformity measure for the $i$-th sample with a supposed label $y$, expressed as $\alpha_i^y$, is determined via the CPSC algorithms, which we will cover in the following subsection. The computation of P-values for a given sample $x_i$ proceeds as outlined below in Eq. (1). In this context, $p_i^y$ denotes the P-value of $x_i$ for a possible label $y$ within the label space. Similarly, $\alpha_j^y$ represents the nonconformity measure for the $j$-th sample, associated with the label $y$ in the calibration set. Lastly, $\alpha_i^y$ stands for the nonconformity measure for a potential label $y$ tied to the $i$-th sample in the proper training set. Here, Laplace smoothing was applied. It should be mentioned that the P-values are calculated on the calibration set with clean data and ground-truth labels.

$$p_i^y = \frac{|\{j = a+1, \ldots, b\} : \alpha_j^y \geq \alpha_i^y| + 1}{b - a + 1} \tag{1}$$

**2.2.2. Nonconformity measure algorithm with shrunken centroids**   In this study, we applied our previously developed CPSC algorithms with the shrunken centroids (SC) as the basis for nonconformity measure because it has shown higher computation efficiency and effectiveness in reliability quantification when compared with the conventional conformal prediction with k-nearest neighbors algorithm (CPKNN) and several other conformal predictors [37], and its effectiveness in quantifying reliability has been validated on both the DILI dataset [7] and the COVID-19 patient ICU admission prediction task [38]. The computation is based on the following steps similar to the steps of shrunken centroids algorithm [39] (assume the dimensionality of feature space is denoted as $D$):

In the feature space, we firstly calculate the class centroids $\bar{x}_m \in \mathbb{R}^D$ for class $1, 2, \ldots, M$) and the overall centroid $\mu$. $C_m$ refers to the set of samples in class $m$, $n_m$ denotes the total number of samples within class $m$ and $x_{jm}$ denotes the $j$-th sample of class $m$.

$$\bar{x}_m = \sum_{x_{jm} \in C_m} \frac{x_{jm}}{n_m} \tag{2}$$

$$\mu = \sum_{i=1}^{n} \frac{x_i}{n} \tag{3}$$

Then, the pooled standard deviation is computed and the contrasts between class centroids and the overall centroid, $d_m \in \mathbb{R}^D$, are normalized using the pooled standard deviation:

$$\sigma^2 = \frac{1}{n-M} \sum_{m=1}^{M} \sum_{x_j \in Cm} \left( x_j - \bar{x}_m \right)^2 \tag{4}$$

$$d_m = \left( \bar{x}_m - \mu \right) / \sigma \tag{5}$$

Next, these contrasts are shrunken towards the overall centroid with a soft threshold symbolized by $\Delta$, which is regarded as a hyperparameter in this study (6):

$$\begin{aligned} d'_m &= \text{sign} \left( \mathrm{d}_m \right) \left( |\mathrm{d}_m| - \Delta \right)_+ \\ h_+ &= \begin{cases} h & h > 0 \\ 0 & h \le 0 \end{cases} \end{aligned} \tag{6}$$

The impact of regularization is governed by the threshold parameter $\Delta$. If the absolute value of $d_{xy}$, denoting the contrast in an $x$-th feature for class $y$, is less than the threshold $\Delta$, it is concluded that the related feature lacks sufficient discriminatory power for classification. Consequently, the shrunken contrast for this attribute is reduced to zero, thus discarding the attribute considered as non-contributory and diminishing the dimensionality of the data. Following this, it becomes possible to recalibrate the class centroids taking into account the shrunken contrasts (7).

$$\bar{x}'_m = \mu + \sigma * d'_m \tag{7}$$

Then, in the feature space, the discriminatory score of a new sample $x^* \in \mathbb{R}^D$ can be compared with shrunken centroids for each class:

$$\delta_m \left( x^* \right) = \log \pi_m - \frac{1}{2} \sum_{k=1}^{D} \frac{\left( x_k^* - \bar{x}'_{km} \right)^2}{\sigma_j^2} \tag{8}$$

where $x_k^*$ denotes the $k$-th feature of the new sample $x^*$ while $\bar{x}'_{km}$ denotes the $k$-th value of the shrunken centroid of class $m$.

The discriminatory score $\delta_m \left( x^* \right)$ quantifies the proximity of a novel sample $\left( x^* \right)$ to the $m$-th shrunken centroid, or in other words, it represents the log probability of $x^*$ being part of class $m$, without normalization. This resulting score is constituted by two components: the initial term, $\log \pi_m$, indicates the prior probability of class $m$, determined by the frequency of samples in class $m$ amidst all observations. The secondary term constitutes the standardized squared distance between the $m$-th centroids and the new sample. Consequently, the log probability of a specified class $m$, devoid of normalization, $P(Y = m|X)$, is influenced by both the initial class distribution and the sample's closeness to varying centroids.

The probability of the new sample $x^*$ from class $k$ can then be modeled with the discriminatory score for all classes:

$$\hat{p} \left( k|x^* \right) = \frac{e^{\delta_k(x^*)/T}}{\sum_{\ell=1}^{K} e^{\delta_\ell(x^*)/T}} \tag{9}$$

In order to obtain a normalized probability distribution (one that spans from 0 to 1 and whose elements sum to 1), the softmax function is employed. This transforms log probabilities (not normalized) from arbitrary real numbers into normalized probabilities. It operates

on $\delta_m(x^*)$ in a manner comparable to its handling of logits in neural networks [40]. Moreover, a scaling factor $T$ (denoted as 'temperature') is implemented to diminish the value of $\delta_m(x^*)$ with the aim of rendering the predicted probability distribution more uniform or "softer", while conserving the relative probability ranks across each class. The concept of temperature hyperparameter $T$ is adopted from the softmax function used in knowledge distillation. This typically results in a more evenly spread distribution across various labels, hence retaining the information of less probable labels while also mitigating overfitting to some degree [41]. Simultaneously, as per (8), the posterior probability as given by $\delta_m(x^*)$ is invariably a negative value. This could lead to the term $e^{\delta_m(x)}$ in the softmax function becoming exceedingly small and potentially causing numerical instability. This risk is particularly pronounced when handling high-dimensional data, as crucial probability information might be lost in these spiky conditional probabilities. By scaling the original $\delta_m(x^*)$ with $T$, the absolute value of the exponential term in the softmax function increases, leading to greater information retention. To mitigate the risks of overfitting and numerical instability, $T$ will be tuned as a hyperparameter in this study.

Finally, we convert the predicted probability to a nonconformity measure in the ICP framework based on (10). Here, we applied a design of the nonconformity measure $\alpha_j^{y_i}$ that has been validated in multiple machine learning applications [7,38,42]:

$$\alpha_j^{y_i} = 0.5 - \frac{\hat{p}\left(y_i \mid x_j\right) - \max \hat{p}_{y! = y_i}\left(y_i \mid x_j\right)}{2} \tag{10}$$

## 2.3. Training data cleaning methods based on inductive conformal prediction

In this study, the cleaning method is visualized in Fig 1. Upon partitioning the entire dataset into training (60%), validation (20%), and test sets (20%), we partition our training data into a proper training set with noise (80% entire training samples) and a well-curated calibration set (20% entire training samples). Here, to simulate the real-world scenarios in biomedical machine learning applications, we let the proper training set represent the large portion of noisy training samples with unknown label qualities, while the calibration set represents the small portion of training data with well-curated labels. As is introduced in the previous subsection, ICP can be used to quantify the conformity of a particular feature-label combination based on the well-curated calibration set. In this study, instead of quantifying the conformity for samples in the test set, we aim to quantify the conformity of the noisy training data in the proper training data with the well-curated calibration set. Then, the wrongly labeled data and outliers can be detected and corrected based on the following rules (shown in step 2 of Fig 1):

- **Wrongly labeled data detection**: if the original label of a training data point has a P-value smaller than another possible label by a large margin (abbreviated as "detection threshold" in the following sections: 0.8/0.5/0.2), the training data point is deemed as wrongly labeled because there is a label that shows better conformity evaluated by the well-curated calibration set;
- **Wrongly labeled data correction**: the label with the highest P-value will be used to replace the original label of the training data point;
- **Outlier detection**: if all of the possible labels are with P-values smaller than 0.1, the training data point is deemed as an outlier since all labels show low conformity to the well-curated calibration set;
- **Outlier correction**: the training data point is removed.

Upon detecting the wrongly labeled data and outliers, corrections are made accordingly. The cleaned proper training set is then combined with the calibration set to be the cleaned training set and used for the downstream prediction tasks (right half section of Fig 1). For the downstream tasks, we used linear discriminant analysis (LDA) and logistic regression (LR) as two representative classifiers. The classifier hyperparameters are fixed with the default values in scikit-learn packages because this study focuses on investigating the influence of the training method process and whether cleaner training data can lead to better classification performances. To investigate how the thresholds to detect the wrongly labeled data affect the performance, we did experiments on three detection thresholds: 0.2, 0.5, and 0.8, and reported all of the results.

For the DILI prediction task, the cleaning method is shown exactly as in Fig 1. For the COVID-19 prediction task, before any classification modeling (both in the training of CPSC and the downstream LDA/LR classifier), the training data have been augmented with synthetic minority oversampling technique (SMOTE) to up-sample the positive cases to balance the two classes [2]. Additionally, for the COVID-19 prediction task and TCGA breast cancer subtype prediction task, Lasso feature selection was performed before any classification modeling to deal with the high feature dimensionality.

The hyperparameters of the CPSC ($\Delta$ and $T$), which is the core algorithm of the cleaning process, as well as the strength of L1 penalty in Lasso feature selection ($C$), were tuned based on the performance on the validation dataset. For the three tasks, the metrics used to feedback the hyperparameter tuning were: classification accuracy for the DILI dataset, the sum of area under the receiver operating characteristic curve (AUROC) and area under the precision recall curve (AUPRC) for the COVID-19 dataset, the sum of classification accuracy and macro-average F1 score for the TCGA dataset. The range of the hyperparameters was: 1) in CPSC: $\Delta$: [0, 0.1, 0.2, 0.3], $T$: [1, 10, 100]; 2) in Lasso feature selection: $C$: [0.05, 0.1, 1, 10].

To simulate different levels of noise in the training data in the real-world applications, we artificially permuted 0 to 80% training labels in the proper training set (increment: 10%). As a result, there were a total of 54 scenarios for each classification task: 3 different detection thresholds, 9 different levels of noise, and 2 different classifiers. 48 scenarios were with the known labeling noise caused by manual label permutation and 6 scenarios were with the original training data.

## 2.4. Model performance evaluation

To evaluate the performance of the training data cleaning method, we investigate both the effectiveness and the dynamics of cleaning processes of the method.

**2.4.1. Effectiveness of training data cleaning**  The effectiveness of the training data cleaning method is quantified by the improvement of classification metrics on the three datasets. On the DILI literature classification task, the classification accuracy was used as the metric considering the balanced classes in the dataset; on the COVID-19 ICU admission prediction task, the AUROC and AUPRC were used as the metric due to the imbalanced dataset with much fewer positive cases; on the TCGA breast cancer subtype prediction task, the multi-class classification accuracy and macro-average F1 score were used as the metric.

**2.4.2. Dynamics of training data cleaning process**  To analyze how the training data cleaning method works and investigate the dynamics of the method, the cleaning processes were visualized. We counted and plotted the total number of corrections for wrongly labeled data detected by the method, the number of wrongly labeled data for each class, as well as the number of outliers detected by the method. Additionally, using the well-curated DILI dataset

as an example, we investigate the correctness of the cleaning processes by counting and plotting the total number of wrongly labeled data detected (same as the total number of corrections), the total number of truly wrongly labeled data, the number of wrongly labeled data after correction. How the cleaning processes variate with the detection thresholds for wrongly labeled data was also investigated.

Finally, we investigate how the cleaning processes variate with the hyperparameters in CPSC ($\Delta$ and $T$), which is the core algorithm of the training data cleaning method and the results are shown in the S4 Fig and S5 Fig.

## 2.5. Statistical tests

To test the robustness of the method, we performed random dataset partitions for 30 times and perform each of the 54 scenarios for 30 repetitive experiments. The mean value and 95% confidence interval (CI) of the classification performance metrics will be reported in the result section. Paired t-tests were performed to test the statistical significance in the metrics with/without training data cleaning.

## 3. Results

### 3.1. Effectiveness of training data cleaning on the drug-induced liver injury literature filtering task

The effectiveness of inductive conformal prediction in training data cleaning was first evaluated on the DILI classification task: to filter publications with drug-induced-liver-injury contents based on the free-text title and abstract. We used two types of word embeddings, and the results are respectively shown in Fig 2 (S2V embeddings) and Fig 3 (W2V embeddings). Each type of text embedding was tested in 54 scenarios, encompassing two classifiers (LR, LDA), three detection thresholds of wrongly labeled data (0.8, 0.5, 0.2) and nine levels of training label permutation from 0 to 80% (increment of 10%). As the percentage of permuted training labels increased, the classification accuracy decreased. In scenarios with no permuted training labels (six scenarios in total), the cleaning process did not yield significant improvements. This outcome can be attributed to the DILI dataset being well curated by FDA experts, suggesting that the original training labels contained minimal noise requiring correction through the cleaning method.

With the S2V embeddings, when the percentage of labels permuted is larger than zero, with the conformal-prediction-based training data cleaning method, the classification accuracy for both LR and LDA models on the test is statistically significantly better in most scenarios on the test set ($p < 0.05$, 47 out of 48 scenarios with permuted training labels). A higher detection threshold for the wrongly labeled data (a stricter strategy to correct the wrongly labeled data from the perspective of the conformal predictor) also leads to more evident accuracy improvement when the noise in the training data is low (percentage of labels permuted lower than 0.3), while a lower detection threshold can lead to more accuracy improvement when the training labels become noisier (percentage of labels permuted larger than 0.4). The largest improvement in test accuracy (in term of absolute value of accuracy increment) was in the scenario with 80% labels permuted and a detection threshold of 0.2 with the LDA classifier: from 0.8120 to 0.9048 (11.4%).

With the W2V embeddings, similar results were shown: the accuracy was significantly better in most scenarios on the test set ($p < 0.05$, 39 out of 48 scenarios with permuted training labels). The effect of the detection threshold for the wrongly labeled data is similar to that shown based on the S2V embedding. What is more evident is that with a detection threshold

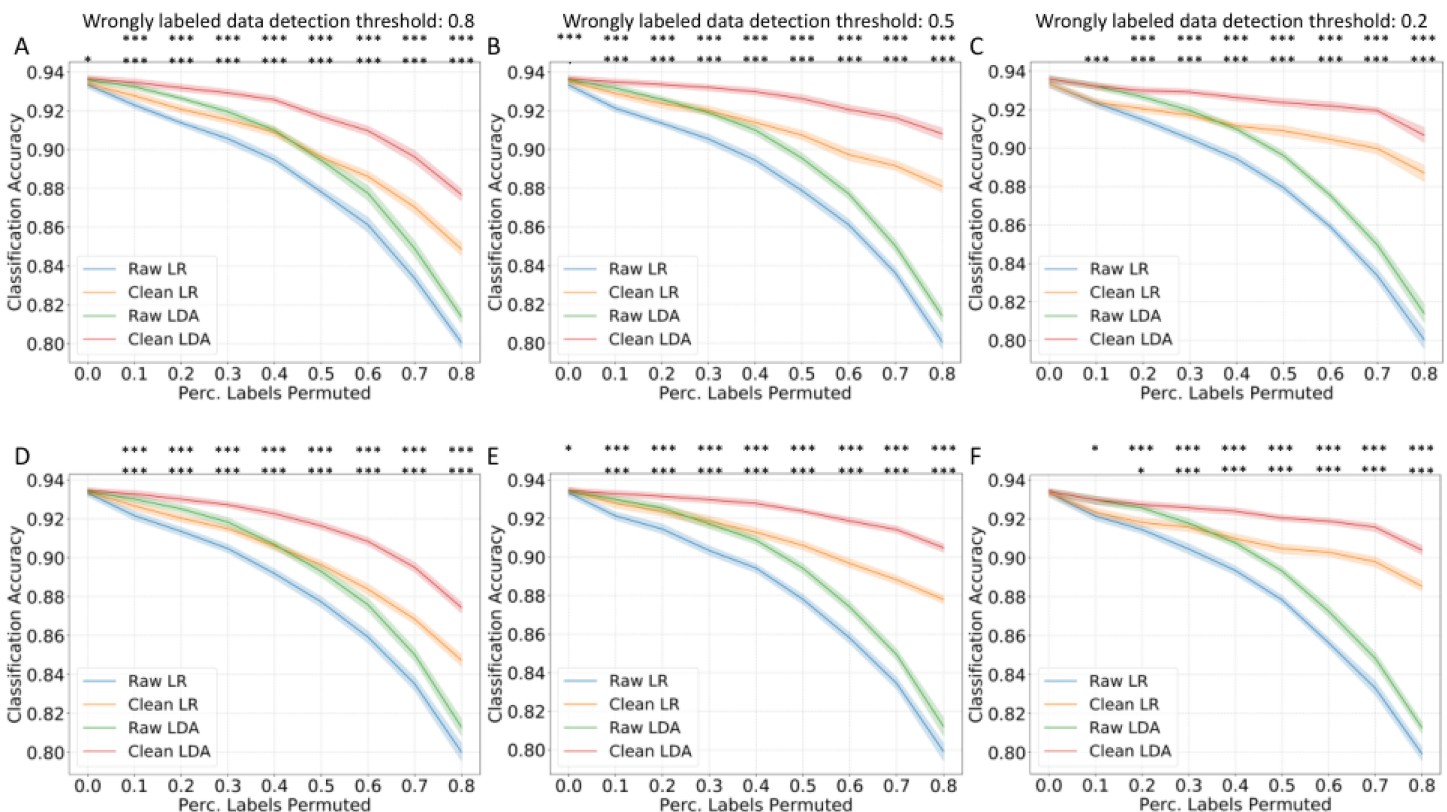

**Fig 2. The model accuracy improvement with training data cleaning in DILI literature classification task based on the S2V embeddings under different percentages of training data label permutation.** The classification accuracy on the validation set (A–C) and on the test set (D–F) with a wrongly labeled data detection threshold of 0.8 (A,D), 0.5 (B,E) and 0.2 (C,F). The mean and 95% confidence intervals are shown. Statistically significant improvement in accuracy has been marked as follows .: $p < 0.1$, *: $p < 0.05$, **: $p < 0.01$, ***: $p < 0.001$; first row: LR models, second row: LDA models.

of 0.2, the cleaning method cannot guarantee model accuracy improvement when the training data is less noisy (i.e., when the percentage of labels permuted is lower than 0.4). The largest improvement in test accuracy (in terms of the absolute value of accuracy increment) was in the scenario with 80% labels permuted and a detection threshold of 0.2 with the LR classifier: from 0.8747 to 0.9019 (3.1%).

## 3.2. Effectiveness of training data cleaning on the COVID-19 ICU admission prediction task

In the COVID-19 ICU admission prediction task: to predict whether a COVID-19 patient admitted to the general ward will be admitted to ICU based on the fusion of radiomics data, clinical data, and laboratory data, the model performance in AUROC and AUPRC are shown in Figs 4 and 5 (as examples with detection thresholds of 0.8 and 0.5 for wrongly labeled data, results for 0.2 detection threshold are shown in S1 Fig) Different from the well-curated DILI dataset, even without artificial label permutation, the original proper training dataset in the COVID-19 task may contain noise after the SMOTE up-sampling for the minority data. The results with a detection threshold of 0.8 (Fig 4) show that even without any artificial label permutation (the percentage of labels permuted is 0), the AUROC and AUPRC can be significantly improved ($p < 0.001$ for LDA on both validation and test sets, $p < 0.01$ for LR on the

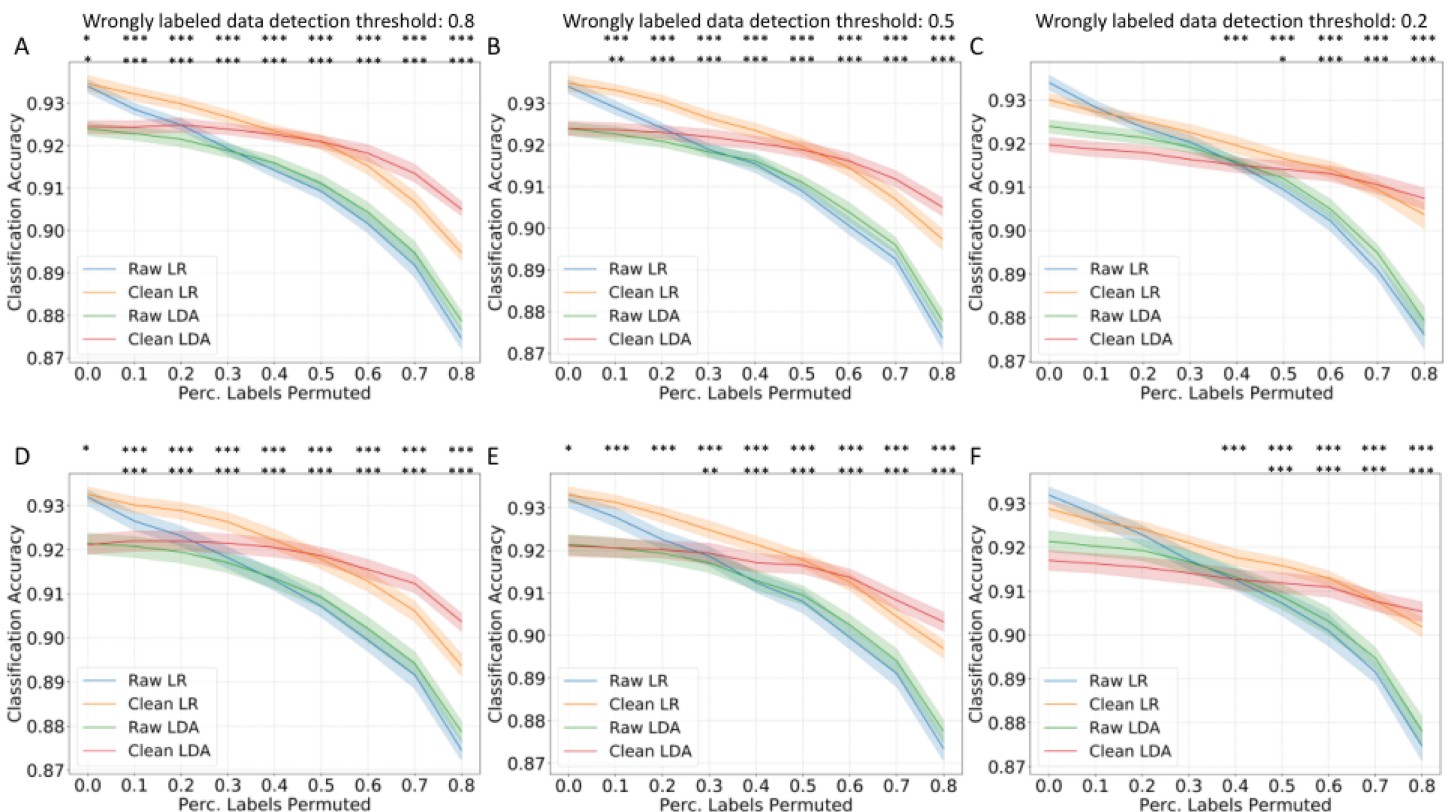

**Fig 3. The model accuracy improvement with training data cleaning in DILI literature classification task based on the W2V embeddings under different percentages of training data label permutation.** The classification accuracy on the validation set (A–C) and on the test set (D–F) with a wrongly labeled data detection threshold of 0.8 (A,D), 0.5 (B,E), and 0.2 (C,F). The mean and 95% confidence intervals are shown. The statistically significant improvement in accuracy has been marked as follows: .: $p < 0.1$, *: $p < 0.05$, **: $p < 0.01$, ***: $p < 0.001$; first row: LR models, second row: LDA models.

validation set). With the increasing percentage of labels permuted, the conformal-prediction-based training data cleaning method shows its effectiveness in improving the AUROC and AUPRC on all scenarios with manual label pollution (n = 48) on the test set ($p < 0.05$ for both AUROC and AUPRC in all scenarios with label permutations). Similar results were shown with a detection threshold of 0.5 (Fig 5): significantly improved AUROC and AUPRC with LDA models without any label permutation ($p < 0.001$), and significantly improved AUROC and AUPRC with label permutation ($p < 0.05$, all scenarios except for the AUPRC with LR with 10% label permutation) on the test set. The largest improvement in test AUROC (in terms of absolute value of increment) was in the scenario with 80% labels permuted and a detection threshold of 0.2 with the LDA classifier: from 0.5967 to 0.7389 (23.8%) while the largest improvement in test AUPRC was in the scenario with 50% labels permuted and a detection threshold of 0.2 with the LDA classifier: from 0.1829 to 0.3106 (69.8%).

### 3.3. Effectiveness of training data cleaning on the breast cancer subtype prediction task

We next evaluated the effectiveness of the reliability-based training data cleaning method in classifying the molecular subtypes of breast cancer using RNA-seq data of the TCGA cohort (n = 1,068 patients) [30]. We considered five subtypes in our analysis, namely LumA (n =

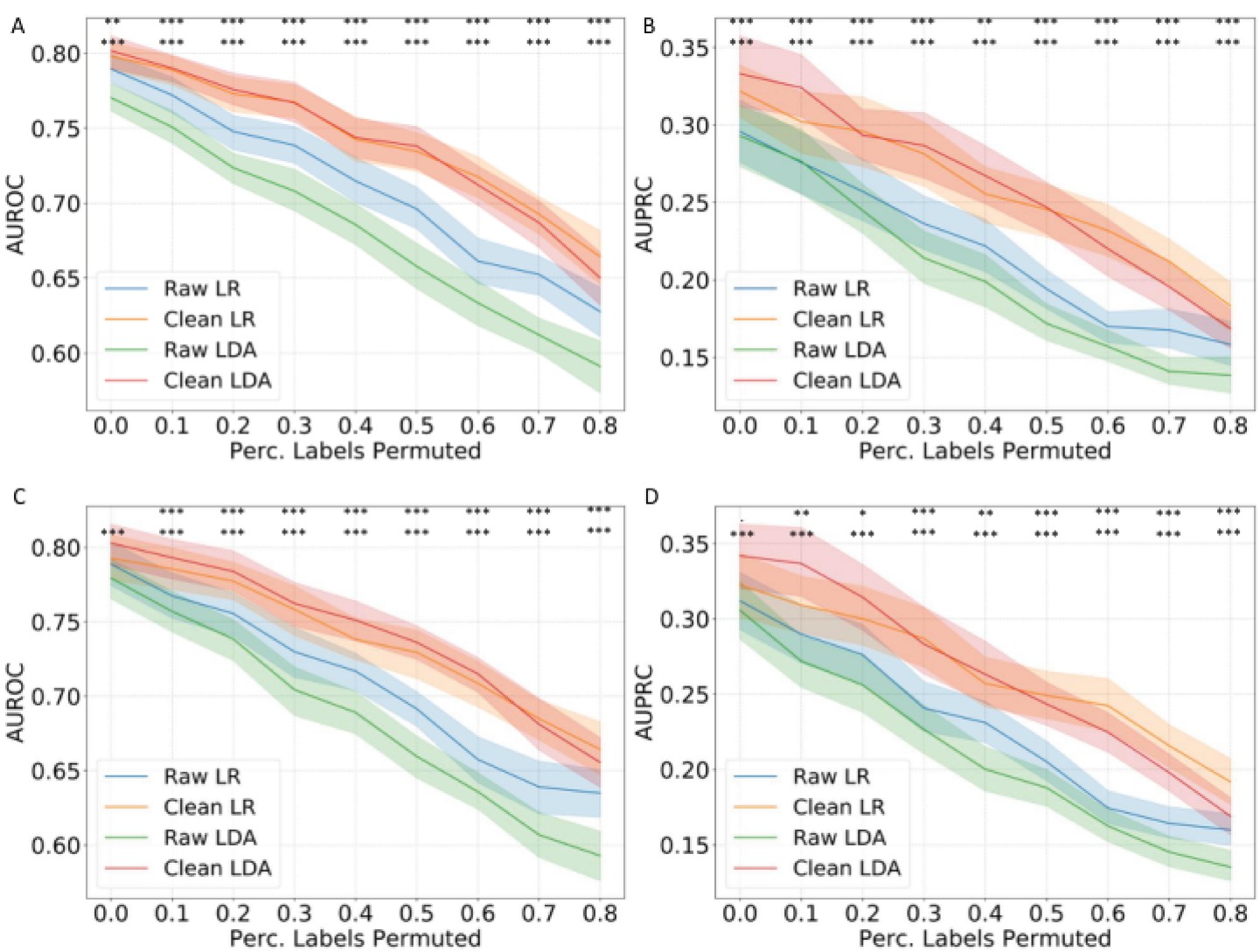

**Fig 4. The model performance in AUROC and AUPRC with training data cleaning in COVID-19 patient ICU admission prediction task under different percentages of training data label permutation.** The AUROC (A) and AUPRC (B) on the validation set, and the AUROC (C) and AUPRC (D) on the test set with a wrongly labeled data detection threshold of 0.8. The mean and 95% confidence intervals are shown. The statistically significant improvement in accuracy has been marked as follows: .: $p < 0.1$, *: $p < 0.05$, **: $p < 0.01$, ***: $p < 0.001$; first row: LR models, second row: LDA models.

555), LumB (n = 210), Basal (n = 184), HER2 (n = 81), and normal-like (n = 38). The accuracy and macro-averaged F1 score are shown in Fig 6 (as an example with detection thresholds of 0.5 for wrongly labeled data, results with detection thresholds of 0.8 and 0.2 are shown in S2 Fig and S3 Fig). In the absence of noise in training labels, the cleaning method did not yield a significant performance improvement, in terms of both classification accuracy and macro-average F1 score. However, when dealing with scenarios containing noisy training labels (n = 48), the data cleaning method leads to a significant improvement in classification accuracy ($p < 0.05$, 47 out of 48 scenarios) and macro-averaged F1 score ($p < 0.05$, 47 out of 48

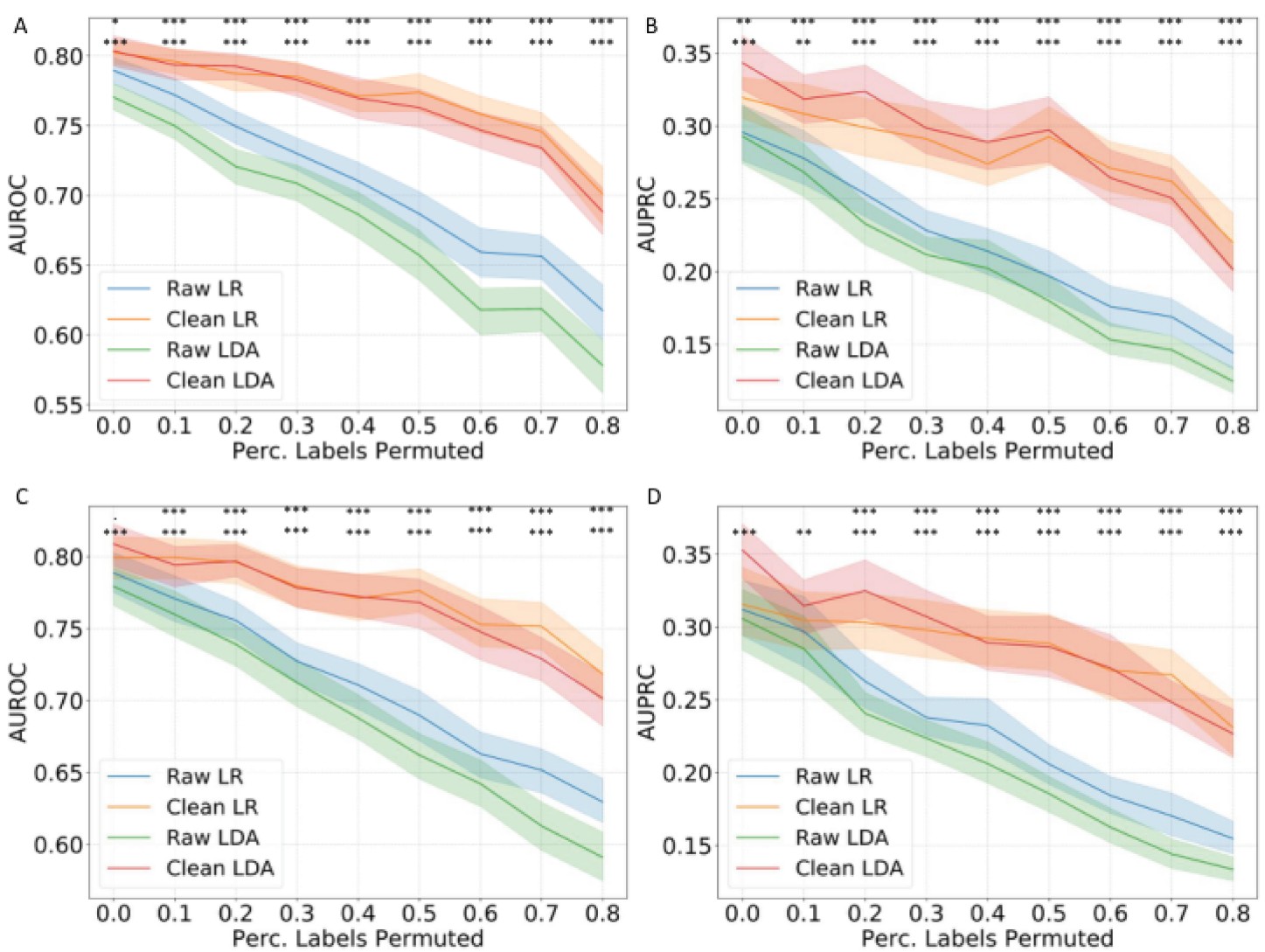

**Fig 5. The model performance in AUROC and AUPRC with training data cleaning in COVID-19 patient ICU admission prediction task under different percentages of training data label permutation.** The AUROC (A) and AUPRC (B) on the validation set, and the AUROC (C) and AUPRC (D) on the test set with a wrongly labeled data detection threshold of 0.5. The mean and 95% confidence intervals are shown. The statistically significant improvement in accuracy has been marked as follows: .: $p < 0.1$, *: $p < 0.05$, **: $p < 0.01$, ***: $p < 0.001$; first row: LR models, second row: LDA models.

scenarios). Similar to the findings from previous tasks, we observed that using lower detection thresholds (less strict rules to detect wrongly labeled data) resulted in better improvements in classification accuracy and macro-averaged F1 score, especially in highly noisy scenarios where over 40% training labels were permuted. The largest improvement in test accuracy (as quantified by the absolute value of increment) was observed when 70% of labels were permuted, and the detection threshold was set to 0.2, using the LDA classifier. In this scenario, accuracy increased by 74.6% (from 0.3508 to 0.6128), while the macro-average F1 score improved by 89.0% (from 0.2672 to 0.5049). Overall, these results demonstrate the effectiveness of our data-cleaning method in improving the classification performance on the TCGA RNA-seq dataset.

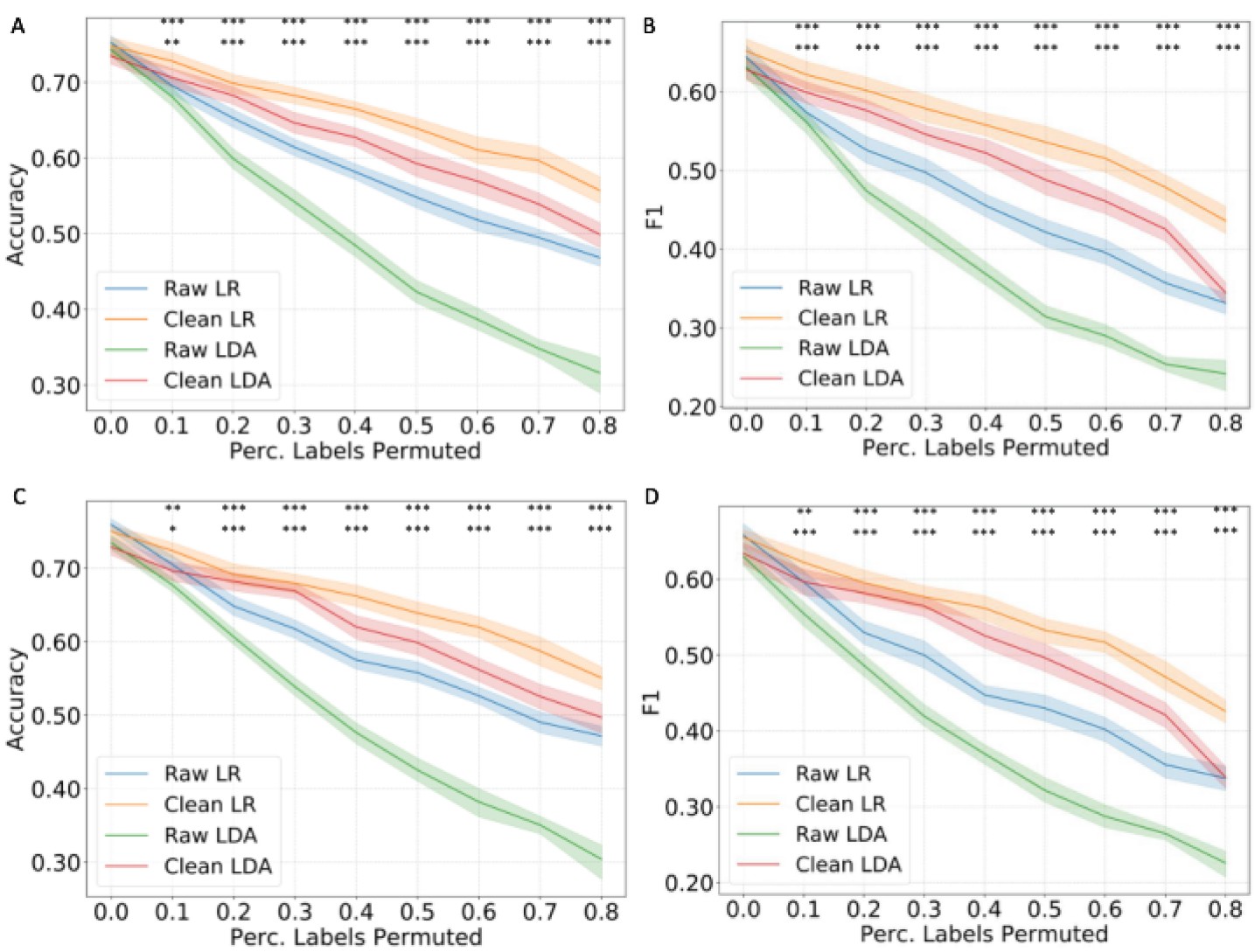

**Fig 6. The model performance in accuracy and F1 score with training data cleaning in the breast cancer subtype prediction task under different percentages of training data label permutation.** The classification accuracy (A) and macro-averaged F1 score (B) on the validation set, and the classification accuracy (C) and macro-averaged F1 score (D) on the test set with a wrongly labeled data detection threshold of 0.5. The mean and 95% confidence intervals are shown. The statistically significant improvement in accuracy has been marked as follows: .: $p < 0.1$, *: $p < 0.05$, **: $p < 0.01$, ***: $p < 0.001$; first row: LR models, second row: LDA models.

## 3.4. Analyses of the cleaning process dynamics based on inductive conformal prediction

To show the detailed processes of the training data cleaning method, the number of wrongly labeled data and outliers are calculated and visualized in Figs 7, 8, 9 and 10. Firstly, we used the DILI W2V dataset as an example to showcase how the cleaning process behaves under different detection thresholds for wrongly labeled data when the hyperparameters of the conformal predictor (CPSC) are fixed (Fig 7, $\Delta = 0.3$ and $T = 100$). As the percentage of labels permuted increases and the proper training data becomes noisier, the number of wrongly labeled data detected by the method increases while it is harder to detect outliers which indicates that the models are less confident to ascertain outliers. Additionally, as the detection

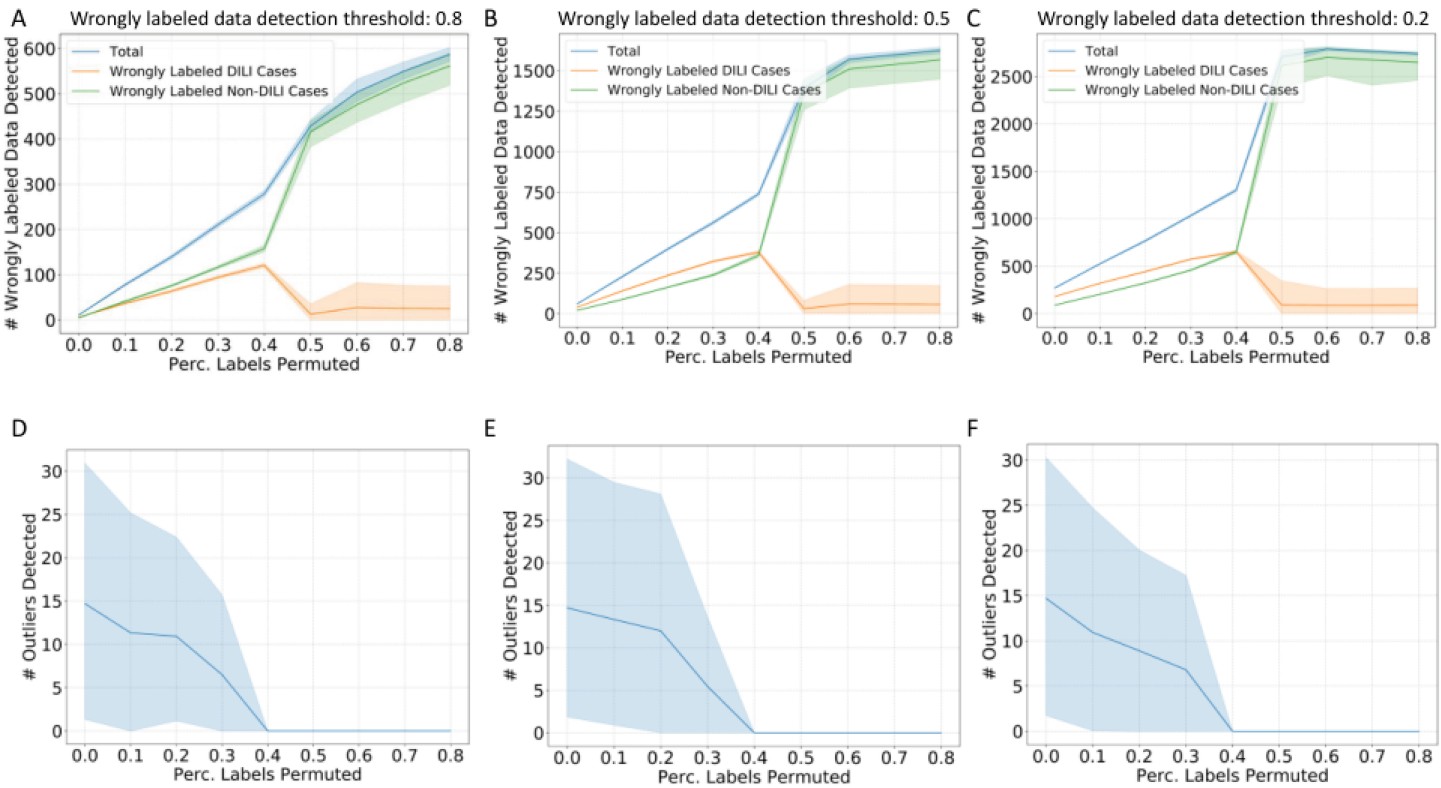

**Fig 7. The number of wrong labels and outliers detected under different percentages of training data label permutation in DILI literature prediction task with W2V embeddings.** The number of wrongly labeled data (A-C) and outliers (D-F) under different detection thresholds of wrongly labeled data: 0.8 (A,D), 0.5 (B,E), 0.2 (C,F). The cleaning process visualization is based on W2V embeddings and fixed hyperparameters for the conformal predictor. Here, total means the total number of wrongly labeled data detected, regardless of labels.

thresholds are set lower, more wrongly labeled data are detected and more label corrections are made by the training data cleaning method. Meanwhile, as the training data grows to be noisier, the detection can be biased toward the wrongly labeled positive cases (DILI-related publications).

Because the DILI data set is well-curated with high-fidelity labels, we were also able to evaluate the correctness of the corrections made by the training data cleaning method by directly comparing the cleaned training data set and the original noisy training data set. Here, we visualized the number of ground-truth wrong labels before and after the training data cleaning processes, as well as the total number of corrections made by the data cleaning method in Fig 8. When the models are stricter in detecting wrongly labeled data (when the detection threshold is set to 0.8), the models are effective in reducing the number of wrongly labeled data after making the corrections. When the detection thresholds are set lower (0.5 and 0.2), when the percentage of labels permuted is lower than 0.5, the corrections are effective in reducing the number of wrongly labeled data. However, as the percentage goes above 0.5 and the training data contain more noise than signals, the data cleaning method can lead to over-correction: after the cleaning process, the number of wrongly labeled data can be even higher.

Additionally, we showed the cleaning process in the COVID-19 patient ICU admission prediction task in Fig 9. Different from Fig 7, we visualized the cleaning process after the

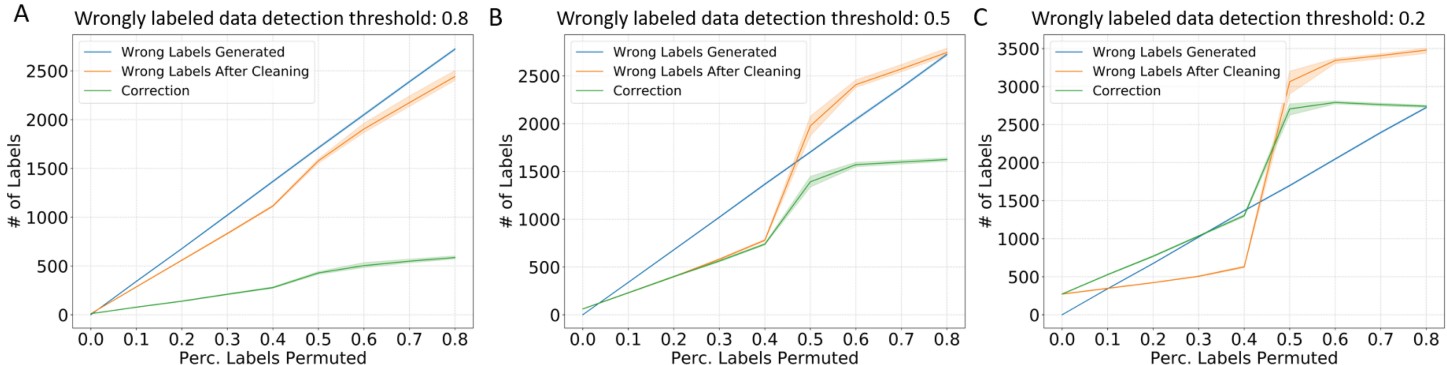

**Fig 8. The number of ground-truth wrong labels before and after training data cleaning under different percentages of training data label permutation in DILI literature prediction task with W2V embeddings.** The number of wrongly labeled data before/after training data cleaning and the number of corrections made under different detection thresholds of wrongly labeled data: 0.8 (A), 0.5 (B), 0.2 (C). The cleaning process visualization is based on W2V embeddings and fixed hyperparameters for the conformal predictor.

CPSC has been optimized via hyperparameter tuning, respectively based on LR and LDA classifiers. Similar to the results on the DILI dataset, a lower detection threshold leads to more wrongly labeled data detected and more corrections, and as the training data become noisier, generally, more corrections for the wrongly labeled data are observed. However, in this task, the wrongly labeled positive cases (patients admitted to the ICU) and wrongly labeled negative cases (patients staying in the general ward) are more balanced. The hyperparameter tuning process leads to certain spikes of wrongly labeled data detected at certain noise levels. In addition, the models are not confident in telling outliers in this task.

For the breast cancer subtype prediction task, similar observations are shown in Fig 10: lower detection thresholds lead to more wrongly labeled data being detected and curated; the number of wrongly labeled data detected for each category corresponds with the prevalence of each subtype.

## 4. Discussion

To address challenges in accurately labeling biomedical data and the limitations of current semi-supervised learning methods, this study proposes a reliability-based data cleaning method using inductive conformal prediction (ICP). In scenarios where only a small portion of well-curated data is available, our method leverages ICP's calibration to improve label reliability within large, noisily labeled datasets. By simulating varying levels of noise across three biomedical tasks—DILI literature filtering, COVID-19 ICU prediction, and breast cancer subtype classification—our method demonstrated effectiveness in enhancing classification performance for both LDA and LR models. Visualizations of the cleaning process highlighted the method's success in identifying mislabeled data, supporting its use in multi-modal biomedical applications. This approach offers a practical solution for researchers needing reliable model training without large, fully annotated datasets.

### 4.1. Unique contribution of this work

Conceptually as a blend of the two mainstream categories of methods to deal with noisy labels introduced in the introduction, our approach addresses the challenge of noisy labels with a novel data cleaning method with inductive conformal prediction with more relaxed

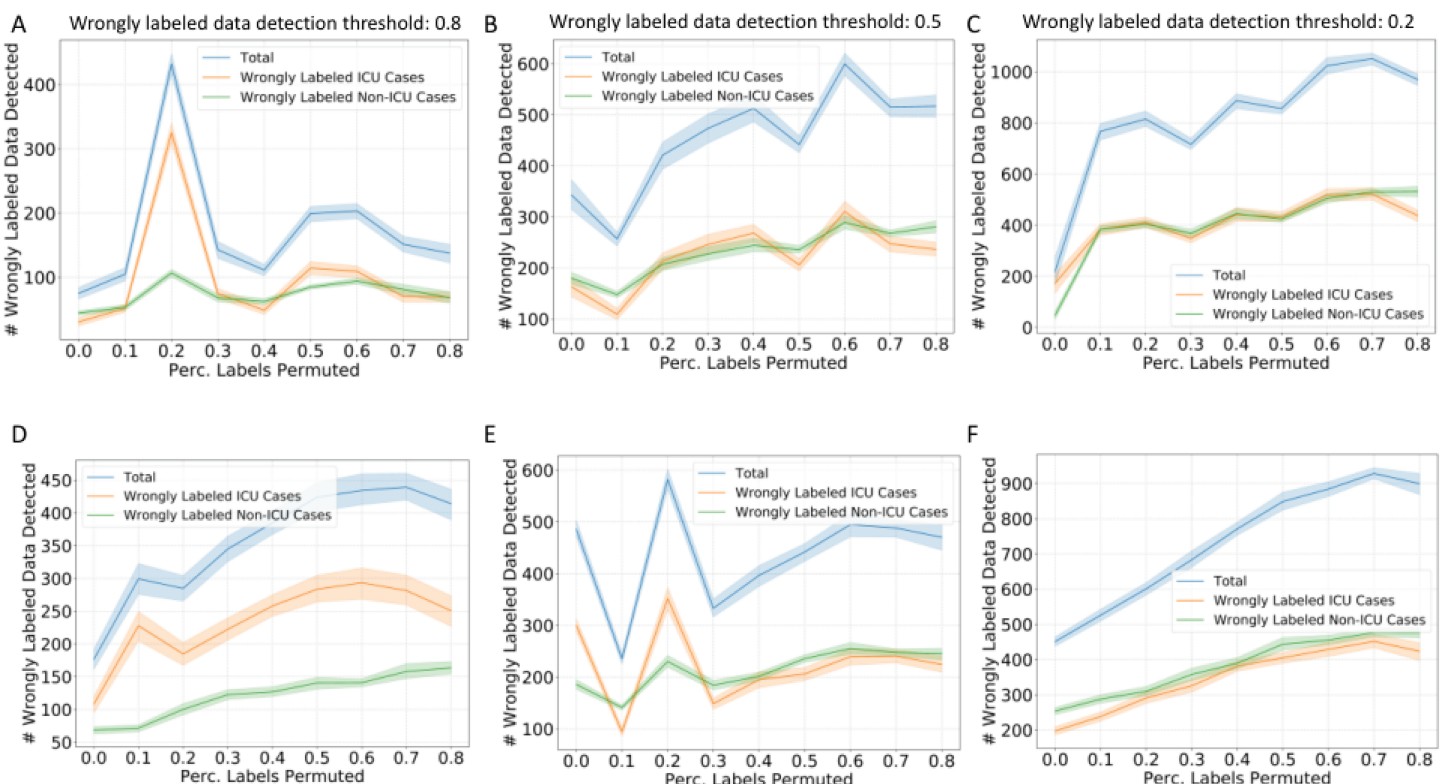

**Fig 9. The number of wrong labels detected under different percentages of training data label permutation in COVID-19 patient ICU admission prediction task.** The number of wrongly labeled data based on LR models (A–C) and LDA models (D–F) under different detection thresholds of wrongly labeled data: 0.8 (A,D), 0.5 (B,E), 0.2 (C,F). The cleaning process visualization is based on optimized hyperparameters for the conformal predictor tuned on the validation dataset for each classifier and each percentage of labels permuted.

assumptions on the models and on the data distribution when compared with the previous approaches [27–29].

Our new method addresses the limitations of the existing solutions: 1) previous semi-supervised learning approaches rely on data distribution assumptions and suffers potential under-utilization of noisy labels [20–22,26]; 2) previous noisy label correction approaches rely on classification modeling assumptions, can easily overfit the clean data and do not consider prediction uncertainty information [14,17,18,25]. Our new method addresses the issue of label noise by introducing a novel approach that combines the strengths of inductive conformal prediction (ICP) with the principles of semi-supervised learning to enhance the quality of noisy training data using a small subset of clean data with high confidence in the labels. This methodology is based on a combination of reliability quantification via ICP and selective data cleaning, aiming to refine the training dataset for improved model robustness. This method enables users to leverage large quantities of noisy labels with the requirement of only a small portion of well-curated training data.

Different from the traditional semi-supervised learning which typically requires strong assumptions of the data distributions for the unlabeled data and labeled data [43], the inductive conformal prediction is based on a weaker assumption of independent and identical distribution. Moreover, this method leverages both well-curated labels and noisy labels with uncertainty. When compared with the previous methods to correct noisy labels, this new

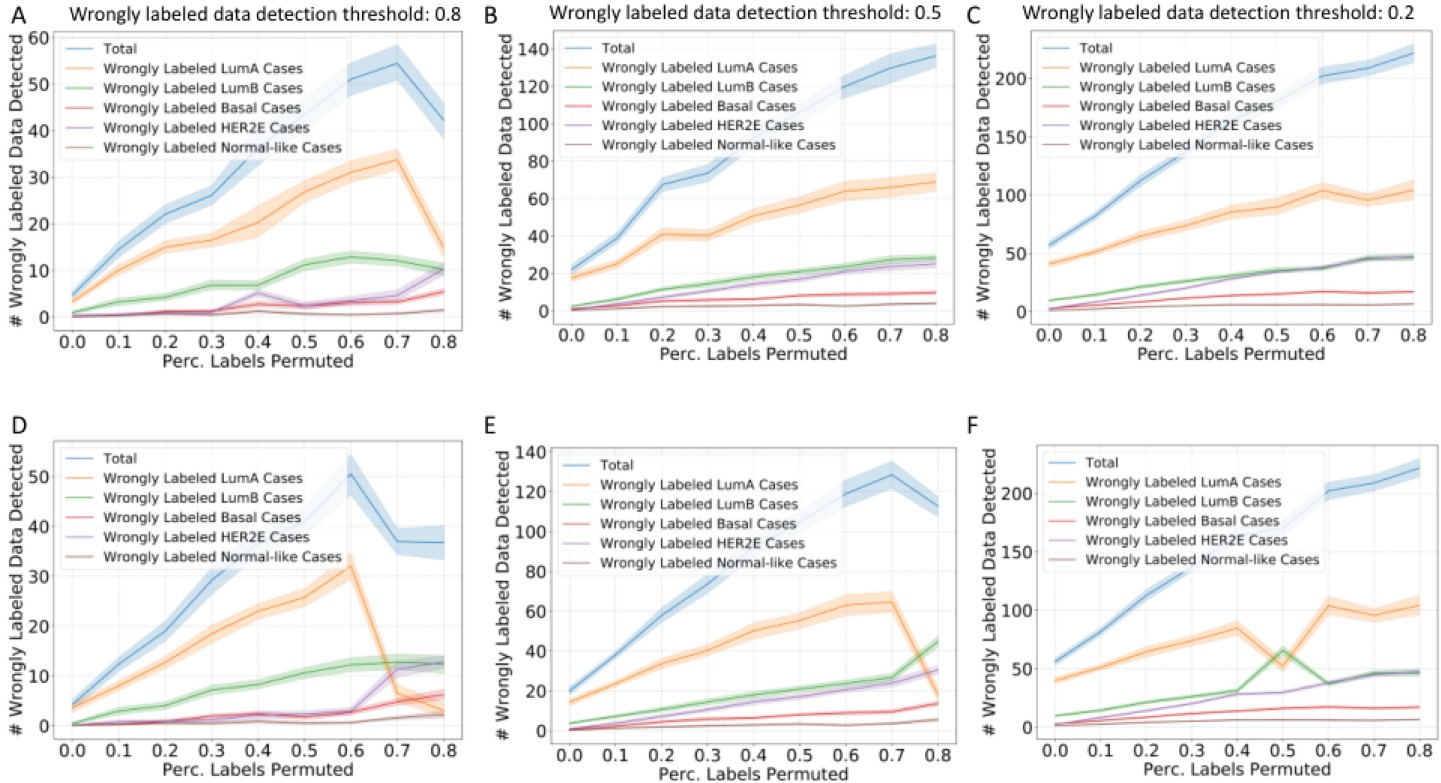

**Fig 10. The number of wrong labels detected under different percentages of training data label permutation in TCGA breast cancer subtype prediction task.** The number of wrongly labeled data based on LR models (A–C) and LDA models (D–F) under different detection thresholds of wrongly labeled data: 0.8 (A,D), 0.5 (B,E), 0.2 (C,F). The cleaning process visualization is based on optimized hyperparameters for the conformal predictor tuned on the validation dataset for each classifier and each percentage of labels permuted.

method relies less on the modeling assumption by using the conformal prediction framework and the statistical calibration, instead of making corrections solely based on the classifier predictions. The new method also mitigates the overfitting to the clean data by using the inductive conformal prediction framework which separates the nonconformity measure and the calibration process. Furthermore, the prediction reliability and uncertainty have been leveraged in this newly proposed work.

## 4.2. Idea creation and relationship with existing research

Although we seek to address the limitations of the existing semi-supervised learning methods, the idea of this work was deeply inspired by the semi-supervised, reliability-based training data augmentation work based on conformal prediction previously proposed [37,42,44]. In previous work, researchers first attach pseudo-labels to the unlabeled data and then leverage the prediction reliability of the pseudo-labels quantified by conformal predictors, to filter these pseudo-labeled data. On the classification tasks, even with domain drifts between the training and test datasets, the reliability-based unlabeled data augmentation framework showed significantly better performances when compared with multiple baseline models: fully supervised learning benchmark, as well as other semi-supervised learning (i.e., label propagation [45] and label spreading [46]) and data augmentation frameworks [44]. These studies

follow the idea of reliability-based unlabeled data augmentation, which inspired us to think in the opposite direction with the idea of reliability-based training data trimming: as we can leverage the reliability quantified by the conformal predictors to filter and add the unlabeled data to benefit the classification modeling process, we should also be able to leverage the reliability to remove the wrongly labeled data in a noisy training set that may confuse the classification modeling process. The rich information given by the nonconformity measure of the conformal predictors enables us to correct the noisy labels in the training data and detect the wrongly labeled data. After implementing this idea, we found that the reliability-based training data cleaning method works well in diverse multi-modal biomedical data sets and classification tasks.

### 4.3. Discussion over model characteristics

The mechanism of the reliability-based training data cleaning method is worthy of further discussion. The method quantifies how well a combination of a training sample's feature with every possible label conforms to the reference distribution of the calibration set. By leveraging and calibrating the nonconformity measure distribution on a small portion of well-curated training data (calibration set), the method can detect whether a training sample may be wrongly labeled after trying out all possible labels that are attached to a training sample's feature. As our method strictly follows the framework of inductive conformal prediction, the training data after cleaning tends not to overfit as both the noisy training data and the clean calibration data set are used at two separate stages: the noisy proper training data were used to fit the conformal predictor while the clean calibration set was used for calibration. The bulk size of the proper training data helps the conformal predictor to learn a general yet fuzzy mapping from the noisy data to the classes while the calibration process better cleans the training data by filtering out the extremely unlikely labels. Instead of using the same dataset to train conformal predictor and calibrate the reliability of predictions, which have been used in some of the previous studies (the non-inductive conformal prediction framework) [7,29,44], the relatively better independence of these two sets based on the inductive conformal prediction framework helps the training data avoid biasing towards the small portion of the calibration set by dissociating the two processes in inductive conformal prediction. As a result, the cleaned training data (the combination of the cleaned proper training set and the calibration set) leads to better classification performances.

Another adaptive feature of the training data cleaning method is the freedom for users to choose the detection threshold of wrongly labeled data, more specifically how much another label's P-value needs to be larger than the current label's P-value to detect a wrongly labeled data point. We have observed that for less noisy training sets (i.e. the percentage of labels permuted is lower than 50%), a higher detection threshold (e.g., 0.8) is more likely to lead to significant improvement in classification performance. On the contrary, for highly noisy proper training sets (percentage of labels permuted over 50%), a lower detection threshold (e.g., 0.2) can lead to better improvement in classification performance. We hypothesize that once the training set gets noisier, less classification information is conveyed in the proper training dataset. Therefore, the CPSC model can be less confident in telling wrongly labeled data. Lowering the detection threshold, in this case, enables more wrongly labeled data to be detected to counteract low confidence in the CPSC model. Therefore, if future users of our method have knowledge of the noise level in the training dataset (e.g., a rough idea of how many labels may be wrongly labeled in a particular dataset), they can choose a high or low detection threshold to help optimize the performance of the training data cleaning method.

### 4.4. Discussion over specific experiment observations

Some observations in the results are also worth further discussion. Firstly, on the DILI task we have observed as the training data grows to be noisier, the detection can bias towards the wrongly labeled positive cases (DILI-related publications) (Fig 7). We assume this is because, in this task, the negatives are by default where a publication is absent of DILI information. This means that the negative samples can be highly heterogeneous: a diverse range of publications associated with vaccine development, optogenetics, epigenetics, transcriptomics, neurological disorders, etc. can be labeled as DILI-negative papers. The DILI-negative samples may not be necessarily a clearly defined class based on contents but grouped because of the absence of DILI information. On the contrary, the positive cases (i.e., the DILI-related publications) are relatively more homogeneous. Therefore, the model tends to be more confident in telling whether a DILI-related publication is wrongly labeled as a DILI-irrelevant publication. In contrast, for the COVID-19 dataset and the TCGA RNA-seq dataset, the classes are more clearly defined.

Secondly, we have observed over-correction in the DILI task in the visualization of the correctness of the training process of the W2V classification task (Fig 8): when the percentage of labels permuted is over 50%, the cleaning can lead to more wrongly labeled data. However, it should be mentioned that although over-correction was observed, the cleaning process still showed effectiveness in improving classification accuracy. We hypothesize that this may potentially be due to that this visualization of cleaning processes is based on fixed hyperparameters with sub-optimal CPSC models, and secondly, potentially these over-corrections may be less influential in the decision boundary determination.

Additionally, although the method was able to detect outliers in the DILI classification task (Fig 7, no outliers were detected in the COVID-19 dataset. We hypothesize that due to that the COVID-19 dataset is noisier than the well-curated DILI dataset because of the SMOTE data augmentation and potential wrong labels in the multi-institute data collection process, the models are less confident in judging outliers in this application.

### 4.5. Limitation

Although this study has proposed an effective training data-cleaning method based on inductive conformal prediction, there are limitations: first, conformal predictions are more generally used in classification tasks to quantify prediction reliability. How much the idea of reliability-based training data cleaning can benefit regression tasks remains unknown. The performance of such a training data cleaning framework needs to be developed and validated in regression tasks to further expand the applicability. Secondly, in this study, we only tested one inductive conformal prediction mechanism, namely, the conformal prediction based on shrunken centroids (CPSC). We chose this framework because previous work showed better efficacy and better efficiency in quantifying the reliability when compared with conformal prediction based on k-nearest neighbors (CPKNN), support vector machines (CPSVM), light gradient-boosting machine (CPLGB) and artificial neural networks (CPANN) [37]. Although the CPSC tends to be more effective and efficient in the reliability quantification process and was more effective in the reliability-based unlabeled data augmentation process, the question of whether our training data cleaning method using other base algorithms works better still remains to be tested. Thirdly, the tasks we tested are with large quantities of samples. With over 500 samples, the partition of the calibration set and proper training set leave both sets not too small and this may have enabled us to better leverage the inductive conformal prediction framework. How well the method works on smaller datasets, and whether we need to use one single clean training set to train conformal predictor as well as perform calibration

(generally abandoning the inductive conformal prediction framework) at the sacrifice of the overfitting risks, need to be further investigated.

## 5. Conclusion

Collecting well-curated training data is challenging while noisy training data generation is easier for biomedical machine-learning applications. However, existing approaches to learn with noisy training data rely heavily on modeling and data distribution assumptions To address this limitation, an inductive-conformal-prediction-based noisy training data cleaning method has been proposed. With a small portion of well-curated training data, our method leverages the conformal-prediction-based reliability to detect the wrongly labeled data and outliers in the large portion of noisy-labeled training data. The effectiveness of this method in improving downstream classification performance is validated on three multi-modal biomedical machine learning classification tasks: detecting drug-induced-liver-injury literature based on free-text title and abstract, predicting ICU admission of COVID-19 patients based on radiomics and electronic health records, subtyping breast cancer based on RNA-seq data. This approach not only enriches the current landscape of semi-supervised learning but also opens new avenues for research into efficient data-cleaning mechanisms that are vital for the progression of machine learning applications in biomedicine.

## Supporting Information

**S1 Text. Cleaning Process Variation with Different Hyperparameters; COVID-19 Data Collection and Preprocessing**
(DOCX)

**S1 Fig. The model performance in AUROC and AUPRC with training data cleaning in COVID-19 ICU admission prediction task under different percentages of training data label permutation.** The AUROC (A) and AUPRC (B) on the validation set, and the AUROC (C) and AUPRC (D) on the test set with a wrongly labeled data detection threshold of 0.2. The mean and 95% confidence intervals are shown. The statistical significant improvement in accuracy has been marked as follows: .: $p < 0.1$, *: $p < 0.05$, **: $p < 0.01$, ***: $p < 0.001$; first row: LR models, second row: LDA models.
(EPS)

**S2 Fig. The model performance in accuracy and F1 score with training data cleaning in the breast cancer subtype prediction task under different percentages of training data label permutation.** The classification accuracy (A) and macro-averaged F1 score (B) on the validation set, and the classification accuracy (C) and macro-averaged F1 score (D) on the test set with a wrongly labeled data detection threshold of 0.8. The mean and 95% confidence intervals are shown. The statistically significant improvement in accuracy has been marked as follows: .: $p < 0.1$, *: $p < 0.05$, **: $p < 0.01$, ***: $p < 0.001$; first row: LR models, second row: LDA models.
(EPS)

**S3 Fig. The model performance in accuracy and F1 score with training data cleaning in the breast cancer subtype prediction task under different percentages of training data label permutation.** The classification accuracy (A) and macro-averaged F1 score (B) on the validation set, and the classification accuracy (C) and macro-averaged F1 score (D) on the test set with a wrongly labeled data detection threshold of 0.2. The mean and 95% confidence intervals are shown. The statistical significant improvement in accuracy has been marked as

follows: .: $p < 0.1$, *: $p < 0.05$, **: $p < 0.01$, ***: $p < 0.001$; first row: LR models, second row: LDA models.
(EPS)

**S4 Fig. The influence of the noise attenuation hyperparameter Δ in the conformal-prediction-based training data cleaning protocol on the number of wrong labels detected in the COVID-19 patient ICU admission prediction task.** The number of wrong labels detected with different Δ: 0 (A), 0.1 (B), 0.2 (C), 0.3 (D). The softmax regularization hyperparameter (T), is set to be 100.
(EPS)

**S5 Fig. The influence of the softmax regularization hyperparameter T in the conformal-prediction-based training data cleaning protocol on the number of wrong labels detected in the COVID-19 patient ICU admission prediction task.** The number of wrong labels detected with different T: 1 (A), 10 (B), 100 (C). The noise attenuation hyperparameter Δ, is set to be 0.1.
(EPS)

## Acknowledgments

We thank the Department of Bioengineering and the Department of Biomedical Data Science, Stanford University for support of this work.

## Author contributions

**Conceptualization:** Xianghao Zhan.

**Data curation:** Xianghao Zhan, Qinmei Xu, Guangming Lu.

**Formal analysis:** Xianghao Zhan.

**Investigation:** Xianghao Zhan, Yuanning Zheng, Olivier Gevaert.

**Methodology:** Xianghao Zhan.

**Project administration:** Olivier Gevaert.

**Resources:** Qinmei Xu, Guangming Lu, Olivier Gevaert.

**Software:** Xianghao Zhan.

**Supervision:** Olivier Gevaert.

**Validation:** Xianghao Zhan.

**Visualization:** Xianghao Zhan.

**Writing – original draft:** Xianghao Zhan, Yuanning Zheng.

**Writing – review & editing:** Xianghao Zhan, Qinmei Xu, Yuanning Zheng, Olivier Gevaert.

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
