## [Decision Letter · Decision Letter 0]

16 Jan 2025

Dear Dr. Gevaert,

We are pleased to inform you that your manuscript 'Reliability-Enhanced Data Cleaning in Biomedical Machine Learning Using Inductive Conformal Prediction' has been provisionally accepted for publication in PLOS Computational Biology.

Best regards,

Anders Wallqvist

Academic Editor

PLOS Computational Biology

Mark Alber

Section Editor

PLOS Computational Biology

Reviewer's Responses to Questions

**Comments to the Authors:**

Reviewer #1: This manuscript describes a method to clean noisy training data for training a classifier. The method was tested on three biomedical datasets of different modalities. The results show that the method improved the classifier's performance in terms of AUROC and AUPRC compared to without applying any cleaning. There are some minor issues that should be addressed before the manuscript can be published:

1. The assumptions behind the method should be elaborated further instead of claiming that it has been "validated" in previous work (line 213, 272). Some assumptions were discussed in Section 4.1 but it's on ICP not on the shrunken centroid used to estimate reliability. It seems that a centroid based method assumes that class distribution is spherical or elliptical.

2. It is not clear why different visualizations were provided for different datasets? Fig 8 is nice to show how many data points the cleaning method corrected compared to the number of wrongly labeled data points. But this is only for DILI not the other two datasets?

3. Some minor English problems in line 568 and 578, Section 4.3. It's models that overfit data not that data are overfitting. Similarly it's models that are bias not data. Data may not be representative to the whole sample space. Those sentences should be revised.

4. Given that even though there are plenty of incorrect corrections by the method, the classification performance did not suffer, this begs a question that the datasets and tasks are so "easy" that even a strong baseline of cleaning can lead to an improvement? Comparison to a strong baseline (say, one of the previous work) may help answer if this is the case.

Reviewer #2: This study introduces a novel approach to data cleaning for machine learning in the biomedical field, utilizing the inductive conformal prediction (ICP) method. The method employs a calibration set to identify and correct mislabeled data and outliers within large, noisy datasets. The approach was validated across three biomedical tasks: the filtering of drug-induced liver injury (DILI) literature, the prediction of ICU admission for patients with SARS-CoV-2, and the classification of breast cancer subtypes using RNA-seq data. The outcomes of this study demonstrate significant enhancements in classification accuracy, AUROC, and F1 scores, even in the presence of substantial noise. This method provides a pragmatic solution for enhancing model performance without the need for extensive data with minimal noise or strict assumptions. Overall, this is a solid paper with great potential. With minor refinements, it could better showcase the novelty and practical implications of the proposed method.

1.While the results are comprehensive, some figures (e.g., Figures 2–6) are dense and could be streamlined. Highlighting key trends or conclusions directly in the figure captions would help readers focus on the most important findings.

2.Although data and code availability is mentioned, providing more detailed replication instructions, such as preprocessing steps or hyperparameter tuning details, would enhance usability for researchers.

3.Certain parts of the text, especially in the abstract and introduction, are overly verbose and could be simplified to make the paper more accessible.

4.While the paper is technical, it should provide brief, reader-friendly explanations of terms like “nonconformity measures” and “calibration sets” to make it accessible to interdisciplinary audiences.

**Have the authors made all data and (if applicable) computational code underlying the findings in their manuscript fully available?**

Reviewer #1: Yes

Reviewer #2: **No: **

PLOS authors have the option to publish the peer review history of their article (what does this mean?). If published, this will include your full peer review and any attached files.

Reviewer #1: No

Reviewer #2: **Yes: **Wenhao Ouyang

---

## [Editor Report · Acceptance letter]

PCOMPBIOL-D-24-02002

Reliability-Enhanced Data Cleaning in Biomedical Machine Learning Using Inductive Conformal Prediction

Dear Dr Gevaert,

I am pleased to inform you that your manuscript has been formally accepted for publication in PLOS Computational Biology. Your manuscript is now with our production department and you will be notified of the publication date in due course.

With kind regards,

Anita Estes
